# Conformational Changes in the BSA-LT4 Complex Induced by the Presence of Vitamins: Spectroscopic Approach and Molecular Docking

**DOI:** 10.3390/ijms23084215

**Published:** 2022-04-11

**Authors:** Nicoleta Cazacu, Claudia G. Chilom, Melinda David, Monica Florescu

**Affiliations:** 1Department of Electricity, Solid Physics and Biophysics, Faculty of Physics, University of Bucharest, Str. Atomistilor No. 405, CP MG-11, Bucuresti-Magurele, 077125 Magurele, Romania; nicoleta.sandu@drd.unibuc.ro (N.C.); claudia.chilom@fizica.unibuc.ro (C.G.C.); 2Department of Fundamental, Prophylactic and Clinical Disciplines, Faculty of Medicine, Transilvania University of Brasov, Str. Universitatii No. 1, Building C, Room CI30, 500068 Brasov, Romania; melinda.dav@gmail.com

**Keywords:** vitamins, BSA-LT4 complex, spectroscopy, molecular docking

## Abstract

Levothyroxine (LT4) is known for its use in various conditions including hypothyroidism. LT4 interaction with serum albumin may be influenced by the presence of vitamins. For this reason, we investigated the effect of vitamin C, vitamin B12, and folic acid on the complex of Bovine Serum Albumin with LT4 (BSA-LT4). UV-Vis spectroscopy was used to monitor the influence of vitamins on the BSA-LT4 complex. Fluorescence spectroscopy revealed a static quenching mechanism of the fluorescence of BSA-LT4 complex by the vitamin C and folic acid and a combined mechanism for vitamin B12. The interaction of vitamin C and folic acid with BSA-LT4 was moderate, while the binding of vitamin B12 was much stronger, extending the storage time of LT4 in blood plasma. Synchronous fluorescence found that the vitamins were closer to the vicinity of Trp than to Tyr and the effect was more pronounced for the binding of vitamin B12. The thermal stability of the BSA-LT4 complex was more evident, but no influence on the stability of BSA-LT4 complex was obtained for vitamin C. Molecular docking studies showed that vitamin C and folic acid bound the same site of the protein, while vitamin B12 bonded to a different site.

## 1. Introduction

Serum albumin is a model protein for ligand binding and transport studies. The structure of serum albumin is influenced by physical and chemical factors such as pH and temperature [1,2]. Serum albumin binds and transports, covalently or reversibly, various ligands, many drugs such as tetracaine [3] and mitomycin C [4]; flavonoids [5] vitamins such as folic acid [6,7], vitamin C [8], vitamin B12 [9]; dyes [10] or nanoparticles [11,12,13]. Also, many hormones are transported by serum albumin, such as steroid hormones [14] or thyroid hormones T3 and T4 [15].

Levothyroxine (LT4) is the chemical equivalent of thyroid hormone T4. LT4 absorption at the cellular level raises clinical issues with major implications for patients’ health. For instance, patients with hypothyroidism experience problems absorbing LT4 due to changes in the body’s internal parameters, such as pH or temperature. One way to improve or correct LT4 absorption may be the intake of vitamins such as vitamin C, folic acid, or vitamin B12. The pathology and diet of patients influences the bioavailability of administered drugs that reach the systemic circulation. Many endogenous and exogenous drugs and dietary supplements such as vitamins are transported by bovine serum albumin (BSA), and can influence the BSA-LT4 interaction.

Vitamin C (ascorbic acid) is one of the essential vitamins due to its role as an anti-aging agent, and because of its protective role against infections, autoimmune diseases, and the development of cancer [16]. There are indications that serum concentrations of thyroid-stimulating hormone (TSH) and free thyroid hormones (T3 and T4) have improved when patients have taken vitamin C [17]. It seems that vitamin C can reduce the adverse effect of heat stress [18]. Also, replacing LT4 with vitamin C prevents oxidative stress in hypothyroid patients [19].

Folic acid is a small water-soluble molecule that is present in artificially enriched foods and pharmaceutical vitamins [20]. Among the benefits of folic acid is the prevention of the development of cancer and the prevention of megaloblastic anemia during pregnancy [21]. Folic acid also plays an important role in DNA synthesis and cell division [22]. It was shown that LT4 decreases serum homocysteine levels much more successfully in the presence of folic acid [23], which means that this complex can decrease the risk of cardiovascular disease by decreasing serum homocysteine levels in hypothyroidism.

Vitamin B12 (cobalamin) is a water-soluble molecule that is synthesized by bacteria from the large intestine of humans [24]. Processes such as DNA synthesis, cellular energy production, and the prevention of megaloblastic anemia [25] are influenced by the concentration of vitamin B12. Studies on patients with multiple sclerosis suggest that there is a direct link between vitamin B12 levels and thyroid hormones, advancing the idea that this complex could bring benefits to patients with multiple sclerosis [24]. Also, a certain percentage of hypothyroid patients have low levels of vitamin B12 [26].

The chemical structures of these vitamins are presented in Figure 1.

In this study, the effect of vitamin C, vitamin B12, and folic acid (FA) on the BSA-LT4 complex were investigated using spectroscopic and molecular docking approaches. The results could help to better understand the interactions that occur between transporter protein BSA and LT4 and BSA-LT4 complex stability in pharmacological applications.

## 2. Results and Discussions

The ability of serum albumin to carry LT4 in the blood plasma may be affected by vitamins. Some vitamins may interfere with the binding of LT4 to BSA. Therefore, in this study, the effects of vitamin C, folic acid, and vitamin B12 as three important vitamins on the binding of LT4 to BSA were investigated.

### 2.1. Structural Changes Highlighted by UV-Absorption

Absorption and fluorescence spectroscopy are complementary methods, providing quantitative information on different analytes when the wavelengths of absorption peaks can be correlated with the changes occurring during an interaction (e.g., type of bonds) [27]. Previously, the BSA-LT4 complex formation was demonstrated [28], thus, in the present work, UV-absorption measurements were used to monitor the influence of vitamins on the BSA-LT4 complex.

Solutions of increasing concentrations of vitamin C ((0–70) μM), FA ((0–70) μM) and B12 ((0–40) μM) were added to the BSA-LT4 complex (3 µM and 15 μM LT4). The concentration of LT4 was chosen to assure the saturation of the protein binding site with LT4. Due to the Trp residue, the maximum of the absorbance of BSA is ~280 nm, while the specific absorbance wavelength for LT4 is ~237 nm. The maximum of BSA absorption shifts towards lower wavelengths (a hypsochromic effect) with increasing vitamin C concentration, while the maximum of LT4 absorption shifts less significantly and bathochromically (Δλ_BSA_ ≅ 11.71 nm and Δλ_LT4_ ≅ 1.74 nm) (see Figure 1A). In the case of FA (see Figure 1B) there is no shift for the BSA absorption maximum, while Δλ_LT4_ ≅ 1.61 nm. However, at ~300 nm, a shoulder appears on the BSA peak for a concentration higher than 20 µM FA, suggesting the appearance of a conjugated system. Both FA and LT4 have an amino moiety. It is known that the BSA-LT4 complex formation occurs via -NH2 groups [28]. Because the structure of the BSA is already complexed with LT4, Figure 1 shows the binding that FA will compete with LT4 in order to bind to the protein active site. A similar scenario occurs in the presence of B12 (see Figure 1C). The BSA absorption peak remains unchanged while, starting with the first concentration of B12, a shoulder at 287 nm appears. Furthermore, the specific absorption peak of B12 (characteristic in water) is clearly visible at 362 nm. In addition, for LT4 absorption, Δλ_LT4_ ≅ 2.80 nm, with a bathochromic shift, suggests a stronger interaction of the BSA-LT4 complex with B12.

### 2.2. The Quenching Mechanism of BSA Fluorescence by Vitamins

Albumin is one of the blood proteins that binds LT4 with moderate affinity [28,29] and carries it in the body. The binding of vitamin C, folic acid, and vitamin B12 to the complex formed between BSA and LT4 was also studied by fluorescence titration.

The fluorescence emission of BSA-LT4 complex monitored at 25 °C decreased with the gradual addition of the vitamins (vitamin C, folic acid, and vitamin B12) in the BSA-LT4 complex (see Figure 2A–C). A similar result was obtained at 35 °C, which suggested that the Trp microenvironment in the LT4 complex protein may be influenced by the addition of vitamins.

There are two types of quenching mechanisms between two molecules in interactions: the static mechanism, through the formation of a complex between the quencher molecule and the fluorophore, and the dynamic mechanism, by a collision process [30]. In order to elucidate the nature of the fluorescence quenching of the BSA-LT4 complex by vitamins, the experimental data, collected at 25 °C and 35 °C, were analyzed according to the Stern-Volmer Eq. (Equation (1)):(1)$$\frac{F_{0}}{F}=1+K_{SV}\left[Q\right]$$
where *F_0_* and *F* are the fluorescence intensities of BSA-LT4 complex in the absence and in the presence of the quencher; [*Q*] is the concentration of the quencher, and *K_SV_* is the quenching constant (Stern-Volmer constant) of the process.

The Stern-Volmer representation was linear for the BSA-LT4/vitamin C and BSA-LT4/folic acid complexes (see Figure 3A). The Stern-Volmer constants of these two complexes were calculated and listed in Table 1. The quenching process was slowly temperature-dependent, but the values of Ksv were lower at 35 °C than at 25 °C, thus the process was not a dynamic one. The same mechanism has previously been found for the binding of folic acid [7] and vitamin C [31] to BSA.

In the case of the complex BSA-LT4/B12 (see Figure 3B), the Stern-Volmer plot showed a positive deviation which suggested that quenching was not only due to collision but may be due to both static and dynamic processes [32,33]. The static quenching constant for the interaction of B12 with BSA-LT4 was determined for the first experimental data that can be fitted with a straight line (Table 1). The bimolecular constant kq was calculated and the values are listed in Table 1. As one can see, the values for kq are larger than the diffusion limit in aqueous solutions, which is 1 × 10^10^ M^−1^ s^−1^ [30]. This is an indication that the interaction between BSA-LT4 and vitamin C, folic acid and vitamin B12 is initiated by a complex formation and not by a collision process. The same mechanism has previously been found for the binding of folic acid [7], vitamin C [31] and B12 [9] to BSA.

### 2.3. Binding Constant and Binding Sites

The strength of the interaction between the BSA-LT4 and the three vitamins can be interpreted in terms of the binding (affinity) constant. Assuming that the static quenching is the main mechanism for the binding of vitamins to BSA-LT4, the number of binding sites and the binding constants were determined according to Eq. Scatchard (Equation (2)):(2)$$\log(\frac{F_{0}}{F}-1)=\log K_{b}+n\log[Q]$$
where *K_b_* is the binding constant, *n* is the number of binding sites, and [*Q*] is the final concentration of the quencher.

The double logarithmic representation of vitamins binding to BSA-LT4 complex was given in Figure 4 by plotting log(*F*_0_*/F* − 1) vs. log([vitamin]). The log*K_b_* was obtained from the intercept and n from the slope of the plot. The values of these parameters were listed in Table 1.

In this study it was considered that one molecule of vitamin bound to the BSA-LT4 complex. The results obtained demonstrate that the binding process was characterized by a moderate interaction of folic acid and vitamin C binding and by a strong interaction for the binding of B12. In a previous study [28], we showed that LT4 binds to BSA with a constant of 5.12 × 10^6^ M^−1^ at 25 °C and 2.59 × 10^6^ M^−1^ at 35 °C. When vitamin B12 binds to the BSA-LT4 complex, the binding constant was 2.45 × 10^7^ M^−1^ at 25 °C and 5.62 × 10^7^ M^−1^ at 35 °C. Thus, vitamin B12 may extend the storage time of bound LT4 in blood plasma. Consequently, the amount of the drug in the target cells is reduced and the maximum effects of LT4 are reduced [34].

The stoichiometry of the binding of vitamin C and folic acid to BSA-LT4 was approximately equal to 1. Therefore, there is one binding site to vitamin C and folic acid with BSA. For the binding of B12 to BSA, the stoichiometry was approximately equal to 1.5. BSA has two Trp residues, Trp134 and Trp212, that are involved in ligand binding. Trp212 is deeply buried in the hydrophobic pocket and Trp134 is more exposed to a hydrophilic environment [35]. Thus, vitamin B12 binds to the hydrophobic pocket located around Trp 212, and a second site may be around Trp 134, but it may be partially occupied.

### 2.4. Thermodynamic Parameters and the Nature of the Binding Forces

Thermodynamic parameters can help to confirm the non-covalent acting forces of an interaction. These parameters can be determined using Equations (3) and (4):(3)$$\ln\frac{K_{b2}}{K_{b1}}=\frac{\Delta H^{0}}{R}\left(\frac{1}{T_{1}}-\frac{1}{T_{2}}\right)$$
(4)$$\Delta G^{0}=-RT\ln K_{b}=\Delta H^{0}-T\Delta S^{0}$$
where *K_b_* is the association constant, *T* is the absolute temperature; *R* is the gas constant (8.314 J K^−1^ mol^−1^).

The values of the thermodynamic parameters of interaction are an indication of the forces that drive the interaction process. For the binding of vitamins to BSA-LT4 complex (see Table 1 and Figure 5) the most enthalpic process is the binding of vitamin B12 to BSA-LT4. Also, for BSA-LT4/vitaminB12 and BSA-LT4/folic acid, ΔH > 0 and ΔS > 0, thus these processes are mainly driven by hydrophobic interaction. For BSA-LT4/vitamin C, ΔH < 0 and ΔS > 0, and this process is driven by electrostatic forces [36].

### 2.5. Conformation Investigation Monitored by Synchronous Fluorescence

Synchronous fluorescence spectroscopy gives specific information for the Tyr or Trp residues when the ∆λ value between the excitation and emission wavelengths is stabilized at either 15 nm (for Tyr) or 60 nm (for Trp) [37]. In order to explore the structural changes in the vicinity of the functional groups of fluorophores in the BSA-LT4 complex by the addition of vitamin C, folic acid, and vitamin B12, synchronous fluorescence spectra were recorded in the presence of vitamins (Figure 6). Successive additions of vitamin C (Figure 6A,B) and folic acid (Figure 6C,D) to the BSA solution led to an insignificant red shift of the maximum emission wavelength of Tyr (∆λ = 15 nm), and for Trp (∆λ = 60 nm) (Figure 6A,B) as well. This revealed that the polarity around the Tyr and Trp residues does not change in the presence of vitamin C and folic acid. In the case of vitamin B12 titration in BSA-LT4 solution (Figure 6E,F), a red shift on 1 nm was observed for the Tyr maximum emission wavelength along with a 3 nm blue shift for Trp. Thus, the microenvironment around Trp became more polar.

By the graphical representation of the F/F_0_ ratio (see Figure 6G), the curve of Δλ = 60 nm is lower than the curve of Δλ = 15 nm, suggesting that Trp is involved in the fluorescent quenching of BSA-LT4 by vitamins. All three vitamins are closer to the vicinity of Trp than to that of Tyr. This effect is more pronounced in the case of vitamin B12 than for vitamin C and folic acid.

### 2.6. The Effect of Vitamins on the Thermal Stability of BSA-LT4 Complex

To investigate the effect of vitamin C, folic acid, and vitamin B12 on the BSA-LT4 complex, fluorescent spectra of the BSA-LT4 complex, in the absence and in the presence of vitamins, were recorded at different temperature values, between 25 °C and 80 °C. The maxima of the fluorescence intensities were represented in Figure 7A. The fraction of the BSA that is thermally denatured was determined according to Equation (5) and is represented in Figure 7B:(5)$$P=\frac{F(T)-F_{N}}{F_{D}-F_{N}}\;\times \;100\%$$
where *P* is the fraction of the denatured protein, *F_N_* and *F_D_* are the fluorescence intensities of the native and denatured states of the protein, and *F*(*T*) is the fluorescence intensity at temperature, *T*.

Considering that the process of thermal denaturation of BSA-LT4 assumes the existence of the model in two states, *N*—native (folded) and *D*—denatured (unfolded), the balance of this process will be established according to the relation (6):(6)$$K_{eq}=\frac{\left[D\right]}{\left[N\right]}$$

The plot of lnK_eq_ vs. 1/T (the van’t Hoff plot) is represented in Figure 7C. The denaturation of the BSA-LT4 complex in the presence of the vitamins is an endothermic process. The slope of the straight line gives the −ΔH_unf_/R value and the Y-intercept gives the ΔS_unf_/R value. The thermodynamic fingerprint of protein stability is given in Table 2.

The values obtained for Tm indicate that the apo-protein denatures faster, and the presence of vitamins induces BSA denaturation at a higher temperature, especially in the presence of vitamin B12 and folic acid. These results suggest that vitamins may induce some stability in the structure of the BSA-LT4 complex.

### 2.7. Molecular Docking Analysis

Molecular docking was used to estimate the best orientation of vitamin C, folic acid, and vitamin B12 at the BSA protein site (see Figure 8). The results obtained suggested that vitamin B12 has the best binding affinity to BSA (Table 3). It also appears to bind to the same site as LT4 as opposed to the other two vitamins.

These results for the BSA-vitamin complexes are correlated with the results obtained in fluorescence, where the binding constant for vitamin B12 appears to increase in the presence of LT4. For the vitamin C and folic acid, other binding sites were obtained compared to LT4, and this is also in accord with the fluorescence results, which showed that the binding constants were not influenced by LT4 very significantly (see Table 1 and Table 3). Figure 8 shows the binding sites for each ligand as well as the amino acids around them positioned at distances of (5.7–15) Å. The main amino acids present near Vitamin B12 are: Asp111, Arg144, Arg196, Arg 458, Arg427, Ala193, Gln403, Glu399, Glu186, Glu519, His145, Lys114, Lys431, Lys523, Pro110, Ser109, Ser192, Ser428, Thr190, Thr518. Around vitamin C are found: Ala290, Arg217, Arg194, Gln195, Glu152, Glu291, His 287, Tyr156, Tyr149, Ser191. Folic acid is surrounded by: Ala193, Arg458, Arg144, Arg435, Asp108, Glu424, Ile455, His145, Lys431, Leu189, Pro110, Ser109, Ser428, Ser 192, Thr190, Tyr451, and Val432.

The driving forces of protein-ligand are hydrophobic, van der Waals or stacking interactions between aromatic amino acids, hydrogen bonds and electrostatic forces [38]. Molecular docking showed that the main forces influencing the binding of vitamin B12 and folic acid to the BSA site were van der Waals forces, conventional hydrogen bonds and hydrophobic forces. In the case of vitamin C, it was observed that the main driving forces were van der Waals and conventional hydrogen bonds. The different results obtained in molecular docking and fluorescence can be explained by the influence that the solvent has on the fluorescence experiments and the condition used in molecular docking. The results obtained in molecular docking show a similar process for the biding of vitamin B12 and folic acid at the BSA biding site, behavior obtained also in fluorescence for BSA-LT4/folic acid and BSA-LT4/vitamin B12. In fluorescence and molecular docking experiments, it was found that the binding of vitamin C to the protein site is done in a different way than the other two vitamins.

## 3. Materials and Methods

### 3.1. Materials

Bovine serum albumin (BSA) (purity over 98%) was purchased from Merk company (Merk KGAA, Darmstadt, Germany), and its concentration was determined spectroscopically, using the standard molar absorption coefficient for Trp and Tyr at 280 nm (ε = 44,000 M^−1^ cm^−1^). Levothyroxine sodium pentahydrate (LT4), with 888.93 g/mol, was purchased from Merk Company (Merk KGAA, Darmstadt, Germany). LT4 was solubilized in dimethyl sulfoxide (DMSO, from Alfa Aesar, Ward Hill, Massachusetts, Statele Unite) to a stock solution of 2.8 mM. Vitamin B12, with 1355.38 g/mol was purchased from ROHT (Karlsruhe, Germany). Vitamin C, with 176.12 g/mol was purchased from Merk Company (Merk KGAA, Darmstadt, Germany), and folic acid with a molecular weight of 441.4 g/mol was purchased from Fluka AG (Buchs, Switzerland). We used N-(2-Hydroxyethyl)piperazine-N′-(2-ethane sulfonic acid) (HEPES ≥ 99.5 %) buffer purchased from the Merck Company (Merk KGAA, Darmstadt, Germany) to prepare the samples. The pH of the samples was established at 7.4 with a saturated NaOH (Merk KGAA, Darmstadt, Germany) solution.

### 3.2. UV-Vis Spectroscopy Measurements

UV spectra were recorded with a UV-Vis FLAME-S spectrometer, Ocean Optics Inc., Largo, FL, USA preconfigured for 200–1050 nm, in a 1 cm × 1 cm quartz cuvette. All measurements were carried out at 24 ± 1 °C.

### 3.3. Fluorescence Measurements

Fluorescence spectra were recorded on a Perkin Elmer MS 55 spectrofluorometer equipped with a 1.0 cm quartz cell. The excitation wavelength for BSA-LT4 complex was 295 nm. The excitation and emission slit widths were fixed at 5.0 nm and 6.0 nm, respectively. The BSA-LT4 complex fluorescence was quenched at 25 °C and 35 °C by successive additions of vitamins: (0–70) μM for vitamin C, (0–70) μM for folic acid, and (0–40) μM for vitamin B12 in the BSA-LT4 complex solution. All spectra were corrected by the inner filter effect.

### 3.4. Synchronous Fluorescence Measurements

Synchronous fluorescence spectra of the BSA-LT4 complex were recorded in the absence and presence of vitamins. The synchronous fluorescence spectra were scanned from 270 nm to 320 nm (Δλ = 15 nm) and from 240 nm to 320 nm (Δλ = 60 nm), respectively.

### 3.5. Molecular Docking

Crystal structures of the BSA (PDB ID: 3V03, [39]) and vitamin B12 [40] were retrieved from RCSB Protein Data Bank [41]. The other structures of ligands were downloaded from the PubChem database, the database of the National Center for Biotechnology Information [42], as follows: LT4 CID = 5819 [43], vitamin C CID = 54,670,067 [44] and folic acid CID = 135,398,658 [45]. For the molecular docking between ligands and BSA, PyRx [46] and UCSF Chimera [47] software were used. The best energetic scoring functions were generated by the AutoDock Vina [48] algorithm. The dimensions of the box were set to (25 × 25 × 25) Å^3^. The diagram of ligand-protein interaction was done using Discovery Studio [49].

## 4. Conclusions

In this study, the binding between the BSA-LT4 complex and three vitamins: vitamin C, folic acid, and vitamin B12, was studied by spectroscopic and molecular docking methods.

Vitamin C and folic acid quenched the fluorescence of the BSA-LT4 complex by a static mechanism, while vitamin B12 interacted with BSA through a combined mechanism that was both static and dynamic. The formed BSA-LT4/vitamin complexes were stabilized by hydrophobic interaction for BSA-LT4/vitamin B12 and BSA-LT4/vitamin C and by electrostatic forces for BSA-LT4/folic acid complex. The highest affinity to the BSA-LT4 complex was found for vitamin B12. In this way, vitamin B12 stabilizes the BSA-LT4 complex, and the immediate effect will be to decrease the concentration of free LT4, (the active form in the cell).

The thermal denaturation study shows that folic acid and vitamin B12 increase the stability of the BSA-LT4 structure more than vitamin C does, the results of which are in accordance with those obtained in UV-Vis spectroscopy.

Molecular docking showed two different binding sites for vitamin C and folic acid and vitamin B12, respectively. In this way, the result obtained in fluorescence which shows that vitamin B12 is closer to the BSA-LT4 site than vitamin C and folic acid is confirmed.

The results of this study, together with further research, should elucidate the physiological importance of vitamins and their influence on LT4 transport and regulation at the molecular level. The results provide important information about the complex behavior of BSA-LT4 in a biological environment. These results could accelerate the implementation of BSA-LT4 as drug delivery systems. Also, these studies are helpful to follow the metabolic, pharmacodynamic, and pharmacokinetic aspects of LT4.

## Data Availability

Not applicable.

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
