# Peer review of "Conformational Changes in the BSA-LT4 Complex Induced by the Presence of Vitamins: Spectroscopic Approach and Molecular Docking"

_ijms, 2022, doi:10.3390/ijms23084215_

Round 1

Reviewer 1 Report

The article is a useful contribution to the understanding of interactions of the BSA-LT4 complex with vitamins by fluorescence spectroscopy and molecular docking. While the experimental and modeling work is of good quality, I have several concerns and comments about the paper, and I suggest that authors give attention to them.

For the purpose of the present work, it is important to ensure that what authors were observing is fluorescence from the BSA-LT4 complex, not fluorescence from BSA alone. The authors were not clear about it, and they should give more details or explanation.  

(In line 89, the meaning of '3 μM and 15 μM LT4' is not clear. Does this means that the concentration of LT4 is five times larger than that of BSA?)

In line 99, 'FA will compete with LT4 for the BSA binding site' seems to be inconsistent with the conclusion from molecular docking, and authors should clarify it. 

In lines 161-164, I am confused about saying that the large K_b value for B12 indicates the strong binding of LT4 to BSA. The K_b value represents the binding between the BSA-LT4 complex and B12, not between BSA-LT4, and how can the K_b value affect the binding of LT4? 

In line 94, the symbols 'ΔλBSA' and 'ΔλLT4' are difficult to read, and 'BSA' and 'LT4' may be better to be written as subscripts. Also, there are many places where the letters should be written as subscripts or superscripts, for example, K_b values in Table 1.  

It seems to me that the numbers in Tables have too many digits. Considering the errors involved in the measurements and analysis, it is hard to believe that they are all significant.   

The sentences in lines 98-99 and line 157 are awkward, and should be corrected. 

Author Response

Comments and Suggestions for Authors

Reviewer #1:

The article is a useful contribution to the understanding of interactions of the BSA-LT4 complex with vitamins by fluorescence spectroscopy and molecular docking. While the experimental and modelling work is of good quality, I have several concerns and comments about the paper, and I suggest that authors give attention to them.

We would like to thank for the Reviewer’s time and detailed attention to read thoroughly our work and for the valuable input, which helped us improve the quality of our manuscript.

  1. For the purpose of the present work, it is important to ensure that what authors were observing is fluorescence from the BSA-LT4 complex, not fluorescence from BSA alone. The authors were not clear about it, and they should give more details or explanation.  

Response: Thank you for the observation. The following details have been added in the article:

  • The fluorescence emission of BSA-LT4 complex monitored at 25 °C decreased by the gradual addition of the vitamins (vitamin C, folic acid, and vitamin B12) (see Fig. 2 A-C). A similar result was obtained at 35 °C, which suggested that the Trp microenvironment in the LT4 complex protein may be influenced by the addition of vitamins.
  • In order to elucidate the nature of the fluorescence quenching of BSA-LT4 by vitamins, the experimental data, collected at 25 °C and 35 °C, were analyzed according to the Stern-Volmer Eq. (Eq.1):
  • where F0 and F are the fluorescence intensities of BSA-LT4 complex in the absence and in the presence of the quencher; [Q] is the concentration of the quencher, and KSV is the quench-ing constant (Stern-Volmer constant) of the process.

  1. In line 89, the meaning of '3 μM and 15 μM LT4' is not clear. Does this means that the concentration of LT4 is five times larger than that of BSA?

Response: The concentration of the ligand (15 μM LT4) in UV-Vis experiment was chosen to observe how the vitamins bind to the protein site (3 μM) when it is almost saturated with LT4.

The text was completed as follows:

Solutions with different concentrations of vitamin C ((0 - 70) μM), FA ((0 - 70) μM) and B12 ((0 - 40) μM) were added to the BSA-LT4 complex (3 µM and 15 μM LT4). The concentration of LT4 was chosen to assure the saturation of the protein binding site with LT4.

  1. In line 99, 'FA will compete with LT4 for the BSA binding site' seems to be inconsistent with the conclusion from molecular docking, and authors should clarify it. 

Response: Thank you for the observation. The following details have been removed from the article:

How this binding site is already occupied by LT4, FA will may compete with LT4 for the BSA binding site.

  1. In lines 161-164, I am confused about saying that the large K_b value for B12 indicates the strong binding of LT4 to BSA. The K_b value represents the binding between the BSA-LT4 complex and B12, not between BSA-LT4, and how can the K_b value affect the binding of LT4? 

Response: Thank you for the observation. We intended to say that vitamin B12 extend the binding time of LT4 to BSA. The following corrections were made:

This is an indication that LT4 was highly bound to BSA in the presence of vitamin B12 and, in this way, vitamin B12 may extend the storage time of LT4 in blood plasma.

Thus, vitamin B12 may extend the storage time of bound LT4 in blood plasma. Consequently, the amount of the drug in the target cells is reduced and the maximum effects of LT4 are reduced [34].

This characterization based on the binding constants has been made in the literature as the following exemple: Ehteshami M., Rasoulzadeh F., Mahboob S., Rashidi M-R. (2013). Characterization of 6-mercaptopurine binding to bovine serum albumin and its displacement from the binding sites by quercetin and rutin, Journal of Luminescence, 135, 164-169

  1. In line 94, the symbols 'ΔλBSA' and 'ΔλLT4' are difficult to read, and 'BSA' and 'LT4' may be better to be written as subscripts. Also, there are many places where the letters should be written as subscripts or superscripts, for example, K_b values in Table 1. 

Response: Thank you for the observation. The following corrections were made in all text, includint Table 1:

  • ΔλBSA @71 nm and ΔλLT4 @ 1.74 nm
  • ΔλLT4 @61 nm
  • ΔλLT4 @80 nm

T

(°C)

KSV x 104/

(M-1)

kq × 1014/

(M-1 s-1)

Kb /

(M-1)

n

ΔG/

(kJ mol-1)

ΔH/

(kJ mol-1)

TΔS/

(kJ mol-1)

BSA-LT4/B12

25

  8.37

12.10

2.45 x 107

1.49

-28.09

63.28

91.38

35

14.47

20.90

5.62 x 107

1.53

-30.44

93.73

BSA-LT4/VC

25

  1.02

1.47

1.12 x 104

1.01

-23.11

-19.29

3.82

35

  0.97

1.40

0.87 x 104

0.99

-23.23

3.94

BSA-LT4/FA

25

  1.06

1.53

0.38 x 104

0.89

-20.43

19.41

39.84

35

  0.84

1.21

0.49 x 104

0.94

-21.76

41.17

  1. It seems to me that the numbers in Tables have too many digits. Considering the errors involved in the measurements and analysis, it is hard to believe that they are all significant.   

Response: Thank you for the observation. In the tables the data has been reduced to two digits after the comma.

  1. The sentences in lines 98-99 and line 157 are awkward, and should be corrected. 

Response: Thank you for the observation. The following corrections were made:

98-99: How this binding site is already occupied by LT4, FA will compete with LT4 for the BSA binding site.

Because the structure of the BSA is already complexed with LT4, in order to bind to the active site of the protein, FA will compete with LT4.

157: One molecule of vitamin bound one molecule of BSA that was already in a com-plex with LT4, and the binding process was characterized by a moderate interaction for folic acid and vitamin C binding and by a strong interaction for B12 binding.

In this study it was considered that one molecule of vitamin bound to the BSA-LT4 complex. The results obtained demonstrate that the binding process was characterized by a moderate interaction of folic acid and vitamin C binding and by a strong interaction for the binding of B12.

Reviewer 2 Report

The paper by Cazacu et al. studies, by means of spectroscopic and computational methods, the effect of the presence of vitamins on a protein-LT4 complex.

The paper is of average interest to the reader of “International Journal of Molecular Sciences “, it is properly written and the conclusions are completely supported by the results.

I’d endorse it for publication on the journal provided that the following minor points are addressed:

  • At line 95, the authors claim that the appearance of shoulder depends on the formation of a conjugated system. Is this really the only possible explanation? A comment on this aspect would be beneficial.
  • Line 137. A “.” is missing before “The static"
  • In order to assess the relevance of the difference in binding affinities listed in Table 3, it'd be important to provide an estimate of the error on these numbers.
  • The authors list the aminoacids located in the vicinity of vitamin B12, vitamin C and folic acid obtained by molecular docking. Can they quantify what “near” means? 

Author Response

Comments and Suggestions for Authors

Reviewer #2:

The paper by Cazacu et al. studies, by means of spectroscopic and computational methods, the effect of the presence of vitamins on a protein-LT4 complex.

The paper is of average interest to the reader of “International Journal of Molecular Sciences “, it is properly written and the conclusions are completely supported by the results.

I’d endorse it for publication on the journal provided that the following minor points are addressed:

We would like to thank for the Reviewer’s time and detailed attention to read thoroughly our work and for the valuable input, which helped us improve the quality of our manuscript.

  1. At line 95, the authors claim that the appearance of shoulder depends on the formation of a conjugated system. Is this really the only possible explanation? A comment on this aspect would be beneficial.

Response: Thank you for the observation. The sentence was removed by the text:

„However, at ~ 300 nm, a shoulder appears on the BSA peak for a concentration higher than 20 µM FA, suggesting the appearance of a conjugated system.”

  1. Line 137. A “.” is missing before “The static"

Response: Thank you for the observation. The corection was made in the text.

  1. In order to assess the relevance of the difference in binding affinities listed in Table 3, it'd be important to provide an estimate of the error on these numbers.

Response: Thank you for the observation. The results obtained in Table 3 were determined using molecular docking. The same results were obtained each time, so no error must be provided.

  1. The authors list the aminoacids located in the vicinity of vitamin B12, vitamin C and folic acid obtained by molecular docking. Can they quantify what “near” means? 

Response: At the order of a few Å (approximately between 5.7-15 Å).

The text was completed as follows:

Fig. 8 shows the binding sites for each ligand as well as the amino acids around them, positioned at distances of (5.7-15) Å